# DiffStyleTS: Diffusion Model for Style Transfer in Time Series

## Abstract

Style transfer combines the content of one signal with the style of another. It supports applications such as data augmentation and scenario simulation, helping machine learning models generalize in data-scarce domains. While style transfer is well developed in vision and language, methods for time series remain limited. We introduce DiffStyleTS, a diffusion-based framework for univariate time series style transfer. DiffStyleTS separates a time series into content and style representations with complementary convolutional encoders and recombines them through an attention-based denoising diffusion process trained without paired style-transfer examples. At inference, the encoders extract content and style from two distinct series, enabling conditional generation of diverse samples that preserve the trajectory of one input while adopting the local dynamics of the other. We demonstrate qualitatively and quantitatively that DiffStyleTS achieves effective style transfer across multiple time-series domains. We further validate its real-world utility by showing that DiffStyleTS-based augmentation improves anomaly detection in a low-data chemical-process setting.

## 1 Introduction

Machine learning models for temporal data often need to generalize beyond the data seen during training. This issue arises in chemical-process engineering (Wagner et al., 2025), energy systems (Deb et al., 2017), finance (Tsay, 2005), and other domains where the same underlying process can appear with different rhythms, noise levels, and local variability. A model trained only on the observed regime may perform well on familiar sequences while failing under plausible shifts in operating conditions. Generative augmentation and scenario simulation address this limitation by creating additional temporal variation that remains consistent with the process of interest (Wen et al., 2021).

*Time series style transfer* (TSST) (El-Laham & Vyetrenko, 2022) provides a direct way to create such variation. As in image (Gatys et al., 2016) and text (Shen et al., 2017) style transfer, the goal is to combine the *content* of one input with the *style* of another. For time-series data, this distinction corresponds to the low-frequency global trajectory and high-frequency local dynamics: content captures the trajectory to be preserved, while style captures short-range variation, volatility, and measurement noise. A useful TSST method should therefore preserve the trajectory of a content series while adapting its local dynamics to a style reference.

This goal is challenging because the separation between content and style is not fixed in temporal signals. The same local fluctuation may be noise in one sequence, a meaningful transient in another, or a change in volatility under a different regime. Existing TSST methods and related approaches manipulate temporal variation by exchanging spectral components, recombining smoothed trends with residuals, or matching neural feature statistics (Takemoto et al., 2019; Tebaldi et al., 2022; El-Laham & Vyetrenko, 2022; Xu et al., 2024). However, these approaches rely on a fixed or prescribed separation between the trajectory and the local variation. As a result, they can misplace, overstate, or suppress the transferred style, and they may not generalize when this separation changes from sample to sample. These limitations motivate a generative TSST model that learns content and style representations from data, while using frequency-based inductive biases to keep the two factors separated, so that it can recombine unseen content and style references across datasets.

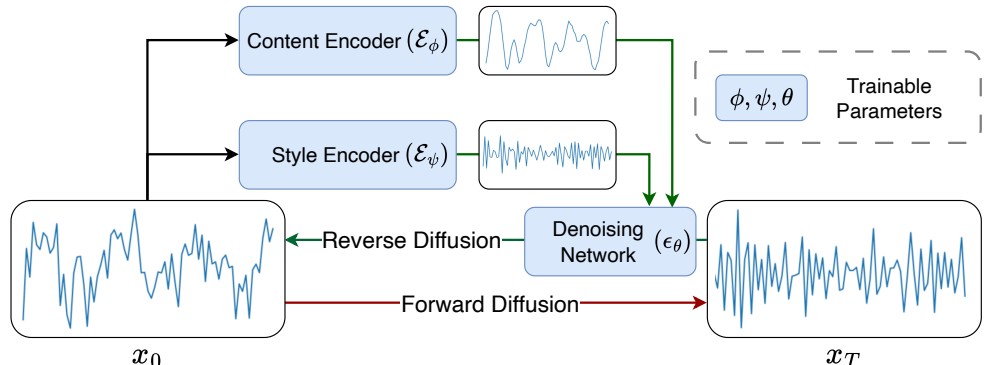

Figure 1: Overview of DiffStyleTS. The input series $x_0$ is gradually corrupted into $x_T$ through an iterative forward diffusion process. Content and style features are extracted by dedicated encoders $\mathcal{E}_\phi$ and $\mathcal{E}_\psi$. In the reverse diffusion process, a denoising network $\epsilon_\theta$ conditioned on these representations progressively reconstructs the input signal. At inference, content and style are drawn from distinct series, enabling the synthesis of new sequences via style transfer.

Diffusion models provide a principled generative backbone for TSST. They learn data distributions through iterative denoising (Ho et al., 2020) and support conditional generation. This has led to strong results in image (Dhariwal & Nichol, 2021), video (Singer et al., 2023), and audio generation (Kong et al., 2021), as well as image style transfer (Wang et al., 2023). For TSST, however, diffusion must be paired with a mechanism that separates content from style. This motivates learning complementary content and style representations from unpaired time series and using them to condition the denoising process, preserving the content trajectory while adapting local dynamics to the style reference.

We introduce **DiffStyleTS**, a diffusion-based framework for univariate time series style transfer that learns separate low-frequency content and high-frequency style streams from unpaired data. Figure 1 gives an overview. DiffStyleTS combines three components: a forward diffusion process that corrupts a time series with noise, convolutional encoders that extract complementary content and style representations, and a conditional denoising network that recombines these representations during reverse diffusion. During training, a single series provides both content and style conditions, making the learning problem self-supervised. During inference, the encoders receive distinct content and style series, and the sampler generates outputs that preserve the content trajectory while adopting the local dynamics of the style reference. We evaluate DiffStyleTS through qualitative examples, a multi-domain benchmark for content preservation, style integration, and realism, and a downstream anomaly-detection task. The results show that DiffStyleTS improves style integration while maintaining realistic outputs, and that DiffStyleTS-based augmentation improves anomaly detection in a low-data chemical-process setting. Our contributions are as follows:

- We formalize TSST through three operational requirements: content preservation, style integration, and realism (Section 3).

- We propose DiffStyleTS, a diffusion-based TSST framework with separate content and style encoders, self-supervised training, and stochastic conditional generation from unpaired data (Section 4).

- We evaluate DiffStyleTS across multiple domains and show gains in qualitative fidelity, quantitative style-transfer behavior, and anomaly detection under data scarcity (Section 5).

## 2  Related Work

DiffStyleTS builds on two lines of work: style transfer methods that separate reusable content from transferable style, and diffusion models that provide conditional, stochastic generation. We review both areas with emphasis on the temporal decomposition required for TSST.

## 2.1 Style Transfer

Style transfer has been studied most extensively in computer vision and natural language processing. In vision, content from one image can be recombined with style from another to synthesize a new image (Gatys et al., 2016; Luan et al., 2017). In text, attributes such as sentiment or formality can be changed while preserving semantic content (Shen et al., 2017; Fu et al., 2018). Time series introduce a different difficulty because observations evolve along an ordered temporal axis, and local fluctuations are meaningful only in relation to the surrounding trajectory (Kantz & Schreiber, 2003). In this setting, content usually denotes the global trajectory, such as long-term trends, while style denotes local dynamics, such as short-range variation, volatility, or noise (El-Laham & Vyetrenko, 2022; Yingzhen & Mandt, 2018).

Closest prior work can be grouped by how it separates and recombines temporal factors. Spectral and multi-resolution methods exchange frequency components (Chaovalit et al., 2011; Yoo et al., 2019); trend–residual methods overlay local deviations on smoothed trajectories (Tebaldi et al., 2022; Xu et al., 2024); and neural style-transfer approaches match feature statistics in a learned or prescribed representation (Gatys et al., 2016; Takemoto et al., 2019; El-Laham & Vyetrenko, 2022). This taxonomy guides our comparisons: the baselines represent frequency exchange, residual recombination, and neural feature matching. MISTI (Hoyez et al., 2025) studies a related multivariate setting in which "style" denotes class-level cross-channel relationships. DiffStyleTS keeps the two-reference TSST objective, but replaces deterministic recombination with conditional diffusion so that the same content and style references can yield diverse plausible outputs under separate content and style controls.

## 2.2 Diffusion Models for Time Series

Diffusion models transform noise into data through a learned denoising process. Denoising Diffusion Probabilistic Models and related samplers have produced high-quality image generation (Song et al., 2020; Dhariwal & Nichol, 2021), and diffusion methods have also been applied to audio (Kong et al., 2021) and text (Gong et al., 2022). Recent work on diffusion guidance further shows that test-time guidance can steer generative behavior without retraining (Pandey et al., 2025). Their appeal for TSST is the combination of conditional control and stochastic generation: a sampler can be guided by reference information while still producing multiple valid outputs.

Recent time-series diffusion models have addressed imputation (Tashiro et al., 2021), forecasting (Rasul et al., 2021), controlled editing (Jing et al., 2024), and extreme value modeling (Galib et al., 2024). TSST poses a different requirement: the model must condition on two complementary references while keeping global trajectory information separate from local dynamics. Since attention-based Transformers (Vaswani et al., 2017) can model dependencies across temporal positions, they provide a suitable denoising backbone for this setting. DiffStyleTS therefore combines separate content and style encoders with a Transformer-based conditional denoiser trained without paired style-transfer examples.

# 3 Problem Setting and Background

We next formalize the TSST objective and provide background on diffusion models. The formalization also defines the requirements used for the evaluation.

## 3.1 Problem Setting

We treat an observed univariate time series as the result of two interacting components. The content component captures the low-frequency global trajectory and long-range structure; the style component captures high-frequency local dynamics and short-range variation. These notions are frequency-related but domain dependent, so the distinction is functional rather than tied to a fixed timescale. Content describes the trajectory to preserve, and style describes the local dynamics to transfer.

The aim of TSST is to generate a time series that preserves the content of one reference while adopting the style of another. Building on style-transfer evaluation in vision and NLP (Gatys et al., 2016; Mir et al., 2019; Ostheimer et al., 2023) and the trend/style distinction in TSST (El-Laham & Vyetrenko, 2022), we formulate

three requirements: content preservation, style integration, and realism. These requirements define what must be controlled at inference time and what must be measured during evaluation.

**Definition 1** (Time Series Style Transfer (TSST)). *Given a content time series $a \in \mathbb{R}^L$ and a style time series $b \in \mathbb{R}^L$, the objective of TSST is to learn a function $f : \mathbb{R}^L \times \mathbb{R}^L \to \mathbb{R}^L$ such that the generated time series $\hat{x} = f(a, b)$ satisfies the following requirements:*

- *(R1) Content Preservation: $\hat{x}$ should preserve the low-frequency trajectory and long-range structure of $a$.*

- *(R2) Style Integration: $\hat{x}$ should reflect the high-frequency local dynamics of $b$, including short-range variation, volatility, and noise.*

- *(R3) Realism: $\hat{x}$ should remain statistically plausible under the relevant time-series data distribution.*

This formulation distinguishes TSST from unconditional generation and standard augmentation. TSST requires two concrete references at inference time: one specifying the trajectory to preserve and one specifying the local dynamics to adopt. The generated sequence must therefore remain aligned with the content reference while expressing style from the second reference.

These requirements are coupled. Strong style integration can distort long-range content, while strict content preservation can suppress the transferred style. Realism adds a further constraint because the output should remain plausible as a time series rather than simply minimizing a content or style distance. The challenge is especially pronounced without paired content–style examples, motivating a generative model that learns the decomposition from unpaired data, conditions on both components, and samples multiple plausible outputs.

## 3.2 Diffusion Models

Diffusion models satisfy this need by learning a structured reverse process that maps Gaussian noise to data. The model is trained through a forward noising process and a learned reverse denoising process, and conditioning variables can be injected into the reverse process to guide generation.

**Forward Process.** A time series $x_0 = x \in \mathbb{R}^L$ is gradually corrupted by Gaussian noise over $T$ steps, yielding $x_0 \to x_1 \to \cdots \to x_T$. Given $x_0$, the distribution at timestep $t$ is

$$q(x_t \mid x_0) = \mathcal{N}\big(x_t \mid \sqrt{\overline{\alpha}_t}x_0, (1 - \overline{\alpha}_t)I\big),$$

where $\overline{\alpha}_t = \prod_{s=1}^{t}(1 - \beta_s)$ determines the noise schedule.

**Reverse Process.** Generation runs the denoising chain $x_T \to x_{T-1} \to \cdots \to x_0$. Since the true reverse transition depends on the unknown clean sample, a neural network approximates it:

$$p_\theta(x_{t-1} \mid x_t) = \mathcal{N}\big(x_{t-1} \mid \mu_\theta(x_t, t), \sigma_t^2 I\big),$$

where $\mu_\theta(x_t, t)$ is predicted by the network and $\sigma_t^2$ is a predefined or learned variance.

**Training Objective.** The denoising model $\epsilon_\theta$ is trained to predict the added noise $\epsilon$ with the objective:

$$\mathcal{L} = \mathbb{E}_{x_0, \epsilon, t}\left[\left\|\epsilon - \epsilon_\theta(\sqrt{\overline{\alpha}_t}x_0 + \sqrt{1 - \overline{\alpha}_t}\epsilon, t)\right\|^2\right].$$

**Conditional Diffusion.** Diffusion models incorporate auxiliary information such as class labels, reference signals, or learned embeddings through the conditional reverse distribution

$$p_\theta(x_{t-1} \mid x_t, c) = \mathcal{N}(x_{t-1} \mid \mu_\theta(x_t, t, c), \sigma_t^2 I),$$

where $c$ shapes the reverse trajectory while retaining stochasticity.

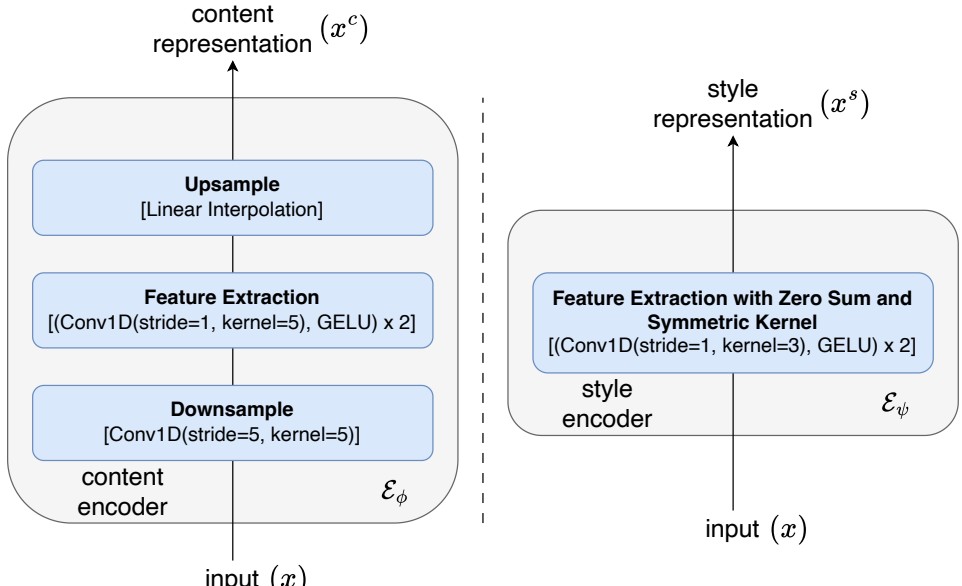

Figure 2: Content and style encoders. Left: The content encoder captures low-frequency content by downsampling the input, processing it at reduced resolution, and upsampling to the original length, preserving the global trajectory while discarding fine details. Right: The style encoder captures high-frequency style using small convolutional filters constrained to avoid constant offsets and phase shifts, ensuring sensitivity to local dynamics. Together, the two encoders separate the global trajectory from local dynamics in the time series.

**Classifier-Free Guidance.** Classifier-free guidance (CFG) (Ho & Salimans, 2021) balances conditioning strength and sample diversity by training the denoiser jointly on conditional and unconditional objectives. During training, conditioning inputs are randomly dropped with probability $p_{\text{drop}}$, enabling the model to learn both conditional $\epsilon_\theta(x_t, t, c)$ and unconditional $\epsilon_\theta(x_t, t)$ denoising functions.

At inference, the two predictions are combined through a guidance scale $s \geq 1$:

$$\epsilon_\theta(x_t, t, c) = \epsilon_\theta(x_t, t) + s \cdot \Big(\epsilon_\theta(x_t, t, c) - \epsilon_\theta(x_t, t)\Big).$$

The modified prediction is used in the sampling update; larger $s$ values increase adherence to the conditioning signal, while smaller values preserve more sampling diversity.

## 4 Methodology

DiffStyleTS represents content and style with separate encoders and recombines them through conditional diffusion. We denote the content series by $a$ and the reference style series by $b$. During training, a single time series $x$ supplies both components ($x = a = b$); during inference, the model receives distinct content and style references. This design lets the model learn from ordinary unpaired time-series data while retaining the two-reference interface required for style transfer.

### 4.1 Extracting Content and Style from a Time Series

DiffStyleTS separates content from style with two complementary encoders, shown in Figure 2. The content encoder emphasizes the low-frequency trajectory and long-range structure, whereas the style encoder emphasizes high-frequency local dynamics and short-range variation. This division gives the denoiser two explicit conditioning streams, one for each temporal factor, and makes it possible to replace either stream at inference time.

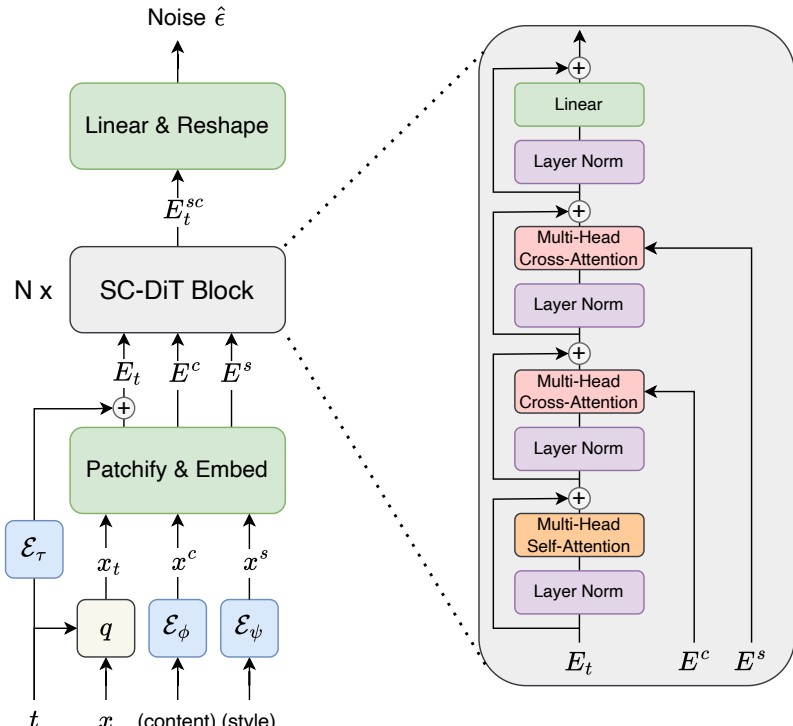

Figure 3: Architecture of the proposed diffusion transformer for time series style transfer. A noisy input $x_t$ is patchified and embedded, combined with a timestep embedding, and processed by a stack of SC-DiT blocks. Each block integrates information from both content and style encoders via cross-attention, while self-attention preserves temporal structure. The network predicts the residual noise $\hat{\epsilon}$ to guide reverse diffusion.

**Content Encoder.** The content encoder $\mathcal{E}_\phi(\cdot)$ maps an input time series $x \in \mathbb{R}^L$ to a content representation $x^c \in \mathbb{R}^L$. It operates in three stages. First, a downsampling module applies a strided convolution with a sufficiently large kernel, acting as a learnable low-pass filter that shortens the sequence and suppresses high-frequency style information. Second, convolutional blocks with large kernels process the reduced-resolution sequence to capture long-range dependencies efficiently. Third, a projection–upsampling stage restores the original length through linear interpolation and projects the result back to a single channel. The resulting representation preserves the global trajectory while filtering out local variation.

**Style Encoder.** The style encoder $\mathcal{E}_\psi(\cdot)$ maps an input time series $x \in \mathbb{R}^L$ to a style representation $x^s \in \mathbb{R}^L$. Unlike the content encoder, it operates at the original resolution so that high-frequency local dynamics are not lost before encoding. A stack of small-kernel convolutional layers captures short-range dependencies, while two constraints keep the filters focused on variation rather than level: zero-mean kernels remove constant offsets, and symmetric kernels enforce linear-phase behavior. Reflection padding preserves the sequence length. Together, these choices make the style encoder a learnable high-pass filter that emphasizes local fluctuations while suppressing global structure.

## 4.2 Denoising Diffusion Transformer for Style Transfer

The denoising network, illustrated in Figure 3, adapts diffusion transformers (DiTs) (Peebles & Xie, 2023) to time series. During the forward diffusion process, Gaussian noise $\epsilon$ progressively corrupts the input series $x$ and produces a noisy sample $x_t = q(x, t) \in \mathbb{R}^L$ at step $t$. The network predicts the added noise $\hat{\epsilon} \in \mathbb{R}^L$ from $x_t$. Cross-attention layers condition this prediction on the content and style features $x^c$ and $x^s$, allowing the reverse process to preserve the content trajectory while injecting style dynamics.

---

**Algorithm 1** Training with Classifier-Free Guidance

---

1: **Input:** dataset $\mathcal{D}$, steps $T$, schedule $\{\beta_t\}$, drop probabilities $p_c, p_s$
2: Initialize $\alpha_t = 1 - \beta_t$ and $\bar{\alpha}_t = \prod_{s=1}^{t} \alpha_s$
3: **repeat**
4:     Sample $x_0 \sim \mathcal{D}$, $t \sim \mathcal{U}\{1, \ldots, T\}$, $\epsilon \sim \mathcal{N}(0, I)$
5:     Forward noising: $x_t = \sqrt{\bar{\alpha}_t}\, x_0 + \sqrt{1 - \bar{\alpha}_t}\, \epsilon$
6:     Compute $x^c = \mathcal{E}_\phi(x_0);\quad x^s = \mathcal{E}_\psi(x_0)$
7:     Apply random dropout: $\tilde{x}^c = x^c$ w.p. $1 - p_c$, else $\varnothing$;$\quad \tilde{x}^s = x^s$ w.p. $1 - p_s$, else $\varnothing$
8:     Take gradient step on: $\nabla_\theta \big\| \epsilon - \epsilon_\theta(x_t, t, \tilde{x}^c, \tilde{x}^s) \big\|^2$
9: **until** convergence

---

**Patch Embedding.** The noisy sample $x_t$, content representation $x^c$, and style representation $x^s$ in $\mathbb{R}^L$ are divided into $n = \frac{L}{p}$ non-overlapping patches of length $p$. A linear projection maps each patch into an embedding space of dimension $h$, producing patch embeddings $E_t, E^c, E^s \in \mathbb{R}^{n \times h}$.

**Time Embedding.** The diffusion step $t$ is encoded as a sinusoidal vector $\tau_t \in \mathbb{R}^h$, following Vaswani et al. (2017). We first define $\frac{h}{2}$ frequencies $\omega_j = \exp\big(-j \cdot \log(10000)/(\frac{h}{2} - 1)\big)$ for $j = 0, \ldots, \frac{h}{2} - 1$. The embedding is then constructed as $\tau_t = \big[\sin(t \cdot \omega_0), \cos(t \cdot \omega_0), \ldots, \sin(t \cdot \omega_{\frac{h}{2}-1}), \cos(t \cdot \omega_{\frac{h}{2}-1})\big]$. The vector $\tau_t$ is added to each patch in $E_t$ to condition the denoiser on the noise level.

**SC-DiT Blocks.** The denoiser is built from stacked *Style-Content Diffusion Transformer* (SC-DiT) blocks. Each block processes $E_t$, $E^c$, and $E^s$ through self-attention and cross-attention, following the standard transformer formulation (Vaswani et al., 2017).

For clarity, we describe one attention head. Queries, keys, and values are defined as $Q = EW^Q$, $K = EW^K$, $V = EW^V$ for input $E$, with learnable $W^Q, W^K, W^V \in \mathbb{R}^{h \times h}$ and $d_k = h$. Self-attention models temporal dependencies within $E_t$:

$$\text{Attn}(E_t) = \text{Softmax}\left(\frac{QK^\top}{\sqrt{d_k}}\right) V.$$

Cross-attention injects the conditioning streams: content information is incorporated from $E^c$ using $Q = E_t W^Q$, $K = E^c W^K$, and $V = E^c W^V$, and style information is incorporated analogously from $E^s$. Each block also includes residual connections, normalization, and feed-forward layers. Stacking several SC-DiT blocks lets the denoiser fuse content and style across time. The output of this stage is denoted $E_t^{sc} \in \mathbb{R}^{n \times h}$.

**Noise Prediction.** The output embeddings $E_t^{sc} \in \mathbb{R}^{n \times h}$ are linearly projected back to the patch dimension $p$. The $n = \frac{L}{p}$ patches are then concatenated to reconstruct a sequence of length $L$, yielding the noise estimate $\hat{\epsilon} \in \mathbb{R}^L$.

**Positional Encodings.** We incorporate relative positional information with ALiBi (Attention with Linear Biases) (Press et al., 2022). Rather than adding positional vectors to the input patches, ALiBi modifies the attention logits with linear biases before the softmax. For positions $i$ and $j$, the bias is $\text{bias}(i, j) = -\alpha|i - j|$, where $\alpha$ is a predefined slope. In self-attention, the bias depends on pairwise patch distances; in cross-attention, it depends on query–key distances. Encoding order directly in the attention scores avoids learned absolute position embeddings and supports extrapolation to longer sequences; Appendix G evaluates this behavior empirically.

### 4.3 Training and Inference

Algorithm 1 summarizes training. For each sample $x_0 \sim \mathcal{D}$, the forward diffusion process produces a noisy input $x_t$. The encoders $\mathcal{E}_\phi$ and $\mathcal{E}_\psi$ extract $x^c$ and $x^s$ from the same clean series $x_0$. This self-supervised setup replaces paired style-transfer examples with the simpler task of reconstructing a noised version of the original series under its own content and style conditions. The denoiser $\epsilon_\theta$ learns to recover the injected

---

**Algorithm 2** Inference with Guidance Combination

---

1: **Input:** content $a$, style $b$, steps $T$, schedule $\{\alpha_t\}$, guidance scales $s_c, s_s$, temperature $\lambda$
2: Initialize $x_T \sim \mathcal{N}(0, I)$
3: Compute $a^c = \mathcal{E}_\phi(a); \quad b^s = \mathcal{E}_\psi(b)$
4: **for** $t = T$ to $1$ **do**
5:     Compute noise predictions: $\epsilon_u = \epsilon_\theta(x_t, t), \;\; \epsilon_c = \epsilon_\theta(x_t, t, a^c), \;\; \epsilon_s = \epsilon_\theta(x_t, t, b^s)$
6:     Combine: $\hat{\epsilon} = \epsilon_u + s_c(\epsilon_c - \epsilon_u) + s_s(\epsilon_s - \epsilon_u)$
7:     Update sample:

$$x_{t-1} = \frac{1}{\sqrt{\alpha_t}}\left(x_t - \frac{1-\alpha_t}{\sqrt{1-\bar{\alpha}_t}}\hat{\epsilon}\right) + \lambda\sigma_t z, \quad z \sim \mathcal{N}(0, I) \text{ if } t > 1, \text{ else } z = 0$$

8: **end for**
9: **Output:** $\hat{x} := x_0$

---

noise $\epsilon$ from $x_t$ conditioned on $(x^c, x^s)$. Classifier-free guidance is enabled by independently dropping the content and style inputs with probabilities $(p_c, p_s)$, so the model learns unconditional, content-conditioned, style-conditioned, and jointly conditioned denoising behavior.

Training reconstructs a series from its own content and style; inference recombines content and style from different references. This recombination uses the same conditional paths learned during training: content and style are encoded separately, attended to through separate cross-attention modules, and dropped independently for CFG. The denoiser therefore learns to use each stream independently and jointly during reverse diffusion, allowing the inference sampler to supply $a^c$ and $b^s$ from different inputs without changing the architecture.

Algorithm 2 summarizes inference. The sampler starts from Gaussian noise $x_T \sim \mathcal{N}(0, I)$ and denoises it over $T$ steps. The content input $a$ and style input $b$ are encoded as $a^c = \mathcal{E}_\phi(a)$ and $b^s = \mathcal{E}_\psi(b)$. At each step, the model predicts unconditional, content-conditioned, and style-conditioned noise estimates; guidance scales $(s_c, s_s)$ combine them into $\hat{\epsilon}$. The temperature $\lambda$ scales the Gaussian noise injected during sampling and therefore controls output diversity. The final sequence $\hat{x}$ preserves the content of $a$ while adopting the style of $b$.

## 5 Experiments

The experiments first evaluate style transfer itself, then test downstream utility in anomaly detection. The style-transfer comparisons use the same baselines and CP/SI/RM metrics for both qualitative and quantitative evaluation. The anomaly-detection study has its own protocol and measures whether DiffStyleTS-generated data improves learning in a low-data chemical-process setting.

### 5.1 Style-Transfer Evaluation Setup

**Datasets and Training.** We train DiffStyleTS on unpaired time series from the training set of Monash Time Series Forecasting Archive (Godahewa et al., 2021), which contains more than 300,000 time series from approximately 30 domains. Training uses 200,000 iterations, 500 diffusion timesteps ($T$), and batches of 256 randomly sampled windows of length 128 on NVIDIA A100 GPUs. Appendix A, Appendix B, and Appendix C provide the architecture, training procedure, and hyperparameters.

**Baselines.** All methods use the same two-reference interface: a content series $a$, a style series $b$, and a transferred output $\hat{x}$. Following Section 2, the baselines represent spectral component exchange, trend–residual recombination, and neural feature-space optimization. This organization makes the comparison interpretable because each baseline asks whether a standard transfer mechanism is sufficient to preserve content, integrate style, and maintain realism. DiffStyleTS uses the same inputs but learns a generative recombination process through conditional diffusion.

*Wavelets* (Yoo et al., 2019; Chaovalit et al., 2011): we use the Haar transform to decompose both content and style series, swap the high-frequency components, and apply the inverse transform to reconstruct the stylized signal.

*Stitching* (Tebaldi et al., 2022; Xu et al., 2024): we smooth the content series to obtain a low-frequency trajectory, treat the residual of the style series as local variation, and add this residual to the smoothed content series.

*Neural Style Transfer (NST)* (Gatys et al., 2016): as an adapted neural baseline, we repurpose the optimization-based vision framework for time series. The generated series $\hat{x}$ is a learnable variable optimized to preserve content and match style statistics. Specifically, we minimize a weighted sum of distances in embedding space:

$$\mathcal{L}(\hat{x}; a, b) = \alpha \left\| f_c(\hat{x}) - f_c(a) \right\|^2 + \beta \left\| f_s(\hat{x}) - f_s(b) \right\|^2, \tag{1}$$

where $f_c(\cdot)$ and $f_s(\cdot)$ denote content and style feature extractors, and $\alpha, \beta$ control the trade-off. For feature extraction, we decompose each series into content and style components using stitching and then embed the components with the MOMENT foundation model (Goswami et al., 2024). This gives the optimization baseline a modern time-series representation while keeping the same content–style interface as the other methods.

**Evaluation Metrics.** Because paired content–style targets are unavailable, we measure the three TSST requirements separately: content preservation on the low-frequency component, style integration on the residual component, and realism in a pretrained time-series representation. To keep evaluation independent of the trained encoders, we use a fixed trend–remainder decomposition (Cleveland et al., 1990) with multi-resolution filtering (Chaovalit et al., 2011). Repeated averaging gives a low-pass trajectory, and the removed residuals capture local dynamics. We decompose the generated series $\hat{x}$, the content reference $a$, and the style reference $b$ with kernels $K = (3, 5, 15)$:

$$\begin{aligned} \hat{x}_{\text{smooth}}^{(i)} &= \text{AvgFilter}_{K_i}\big(\hat{x}_{\text{smooth}}^{(i-1)}\big), \quad i = 1, \ldots, |K|, \\ \hat{x}_{\text{res}}^{(i)} &= \hat{x}_{\text{smooth}}^{(i-1)} - \hat{x}_{\text{smooth}}^{(i)}, \end{aligned} \tag{2}$$

where $\text{AvgFilter}_L(\hat{x}) = \hat{x} * \frac{1}{L}\mathbb{1}_L$, $\hat{x}_{\text{smooth}}^{(0)} = \hat{x}$, the final smoothed signal $C(\hat{x}) = \hat{x}_{\text{smooth}}^{(|K|)}$ represents content, and the aggregated residuals $S(\hat{x}) = \sum_{i=1}^{|K|} \hat{x}_{\text{res}}^{(i)}$ represent style. The same fixed decomposition is applied to every method.

*Content Preservation (CP).* Content Preservation compares the smoothed generated series with the smoothed content reference:

$$\text{CP}(\hat{x}, a) = \frac{1}{L} \sum_{t=1}^{L} (C(\hat{x})_t - C(a)_t)^2. \tag{3}$$

This penalizes low-frequency drift or deformation relative to the content input while being less sensitive to local fluctuations that style transfer may change.

*Style Integration (SI).* Style Integration compares the generated residual with the style-reference residual:

$$\text{SI}(\hat{x}, b) = \frac{1}{L} \sum_{t=1}^{L} (S(\hat{x})_t - S(b)_t)^2. \tag{4}$$

This focuses the comparison on local dynamics, including short-range variation, volatility, and noise, after the global trajectory has been removed.

*Realism (RM).* Realism adds a representation-space plausibility check, following the common practice of evaluating generated samples in a pretrained feature space (Heusel et al., 2017). We use MOMENT, a frozen time-series foundation model (Goswami et al., 2024), and compare both decomposed components in its embedding space:

$$\text{RM}(\hat{x}, a, b) = \tfrac{1}{2}\Big[ \text{MSE}\big(f(C(\hat{x})), f(C(a))\big) + \text{MSE}\big(f(S(\hat{x})), f(S(b))\big) \Big], \tag{5}$$

Table 1: Quantitative evaluation of time series style transfer methods. DiffStyleTS achieves the best overall score and is best or competitive across most datasets. Reported are overall scores (mean of CP, SI, RM per sample). Lower is better.

| Dataset | Stitching | Wavelets | NST | DiffStyleTS |
|---|---|---|---|---|
| Electricity Demand | $0.07 \pm 0.01$ | $0.08 \pm 0.01$ | $0.12 \pm 0.01$ | $\mathbf{0.07 \pm 0.01}$ |
| Bitcoin | $0.06 \pm 0.00$ | $0.07 \pm 0.00$ | $0.11 \pm 0.00$ | $\mathbf{0.05 \pm 0.00}$ |
| Covid Deaths | $0.08 \pm 0.00$ | $0.09 \pm 0.01$ | $0.13 \pm 0.01$ | $\mathbf{0.07 \pm 0.00}$ |
| FRED-MD | $0.04 \pm 0.00$ | $0.05 \pm 0.00$ | $0.09 \pm 0.00$ | $\mathbf{0.04 \pm 0.00}$ |
| Kaggle Web Traffic | $0.08 \pm 0.00$ | $0.09 \pm 0.00$ | $0.13 \pm 0.00$ | $\mathbf{0.05 \pm 0.00}$ |
| KDD Cup 2018 | $0.11 \pm 0.00$ | $0.13 \pm 0.00$ | $0.20 \pm 0.01$ | $\mathbf{0.08 \pm 0.00}$ |
| London Smart Meters | $\mathbf{0.16 \pm 0.00}$ | $0.17 \pm 0.00$ | $0.29 \pm 0.00$ | $0.19 \pm 0.00$ |
| NN5 Daily | $0.19 \pm 0.01$ | $0.26 \pm 0.01$ | $0.45 \pm 0.01$ | $\mathbf{0.12 \pm 0.00}$ |
| Oikolab Weather | $0.14 \pm 0.01$ | $0.19 \pm 0.01$ | $0.24 \pm 0.01$ | $\mathbf{0.12 \pm 0.01}$ |
| Pedestrian Counts | $0.16 \pm 0.00$ | $0.19 \pm 0.00$ | $0.25 \pm 0.00$ | $\mathbf{0.10 \pm 0.00}$ |
| Rideshare | $0.16 \pm 0.01$ | $0.18 \pm 0.01$ | $0.25 \pm 0.02$ | $\mathbf{0.14 \pm 0.01}$ |
| Saugeenday | $0.09 \pm 0.00$ | $0.10 \pm 0.00$ | $0.15 \pm 0.00$ | $\mathbf{0.06 \pm 0.00}$ |
| Solar (10min) | $0.07 \pm 0.00$ | $0.08 \pm 0.01$ | $0.12 \pm 0.01$ | $\mathbf{0.07 \pm 0.01}$ |
| Solar (4sec) | $\mathbf{0.05 \pm 0.00}$ | $0.05 \pm 0.00$ | $0.09 \pm 0.00$ | $0.05 \pm 0.00$ |
| Sunspot | $0.18 \pm 0.01$ | $0.22 \pm 0.01$ | $0.30 \pm 0.02$ | $\mathbf{0.13 \pm 0.01}$ |
| Temperature (rain) | $\mathbf{0.17 \pm 0.00}$ | $0.26 \pm 0.00$ | $0.42 \pm 0.00$ | $0.17 \pm 0.01$ |
| Tourism (monthly) | $0.11 \pm 0.00$ | $0.14 \pm 0.00$ | $0.29 \pm 0.00$ | $\mathbf{0.10 \pm 0.00}$ |
| Traffic (hourly) | $0.24 \pm 0.01$ | $0.28 \pm 0.01$ | $0.33 \pm 0.01$ | $\mathbf{0.20 \pm 0.01}$ |
| US Births | $0.20 \pm 0.00$ | $0.28 \pm 0.00$ | $0.40 \pm 0.00$ | $\mathbf{0.16 \pm 0.00}$ |
| Vehicle Trips | $0.19 \pm 0.00$ | $0.26 \pm 0.00$ | $0.38 \pm 0.00$ | $\mathbf{0.12 \pm 0.00}$ |
| Weather | $0.12 \pm 0.00$ | $0.16 \pm 0.00$ | $0.23 \pm 0.00$ | $\mathbf{0.10 \pm 0.00}$ |
| Wind (4sec) | $0.05 \pm 0.00$ | $0.06 \pm 0.00$ | $0.10 \pm 0.00$ | $\mathbf{0.05 \pm 0.00}$ |
| Wind Farms (min) | $\mathbf{0.05 \pm 0.00}$ | $0.09 \pm 0.00$ | $0.08 \pm 0.00$ | $0.06 \pm 0.00$ |
| Overall | $0.12 \pm 0.01$ | $0.15 \pm 0.01$ | $0.22 \pm 0.02$ | $\mathbf{0.10 \pm 0.01}$ |

where $f(\cdot)$ denotes the MOMENT embedding function. RM measures whether the generated content and style components remain close to their references in a learned time-series representation. Together, these three scores form a task-specific benchmark for TSST, where many outputs can validly combine the same content and style references. Lower values are better for all metrics. Appendix D provides the formal metric definitions.

## 5.2 Quantitative Style-Transfer Benchmark

For the quantitative benchmark, content references come from test splits of the Monash archive (Godahewa et al., 2021). Style references come from five HIVE-COTE datasets (Middlehurst et al., 2021) that are excluded from training, ensuring that evaluation uses unseen styles. Table 1 reports the overall score, computed as the mean of CP, SI, and RM per sample (lower is better), across all evaluated datasets. The same decomposition and embedding metrics are applied to every method. DiffStyleTS obtains the best overall score and is best or competitive on most Monash datasets, indicating that the learned recombination generalizes to unseen style references. Appendix F reports the CP/SI/RM breakdown: deterministic baselines often achieve near-zero CP because they reuse parts of the content input, while DiffStyleTS accepts a modest CP cost in exchange for stronger SI.

DiffStyleTS also matches or surpasses competing methods on RM, indicating that its outputs remain close to realistic content and style representations in the chosen embedding space. Its largest gains appear on SI, which is computed from the fixed residual decomposition and is independent of the pretrained embedding. In datasets such as London Smart Meters, Temperature, and Wind Farms, stitching or wavelets achieve lower overall scores because direct manipulation is sufficient. DiffStyleTS nevertheless adds stochastic generation

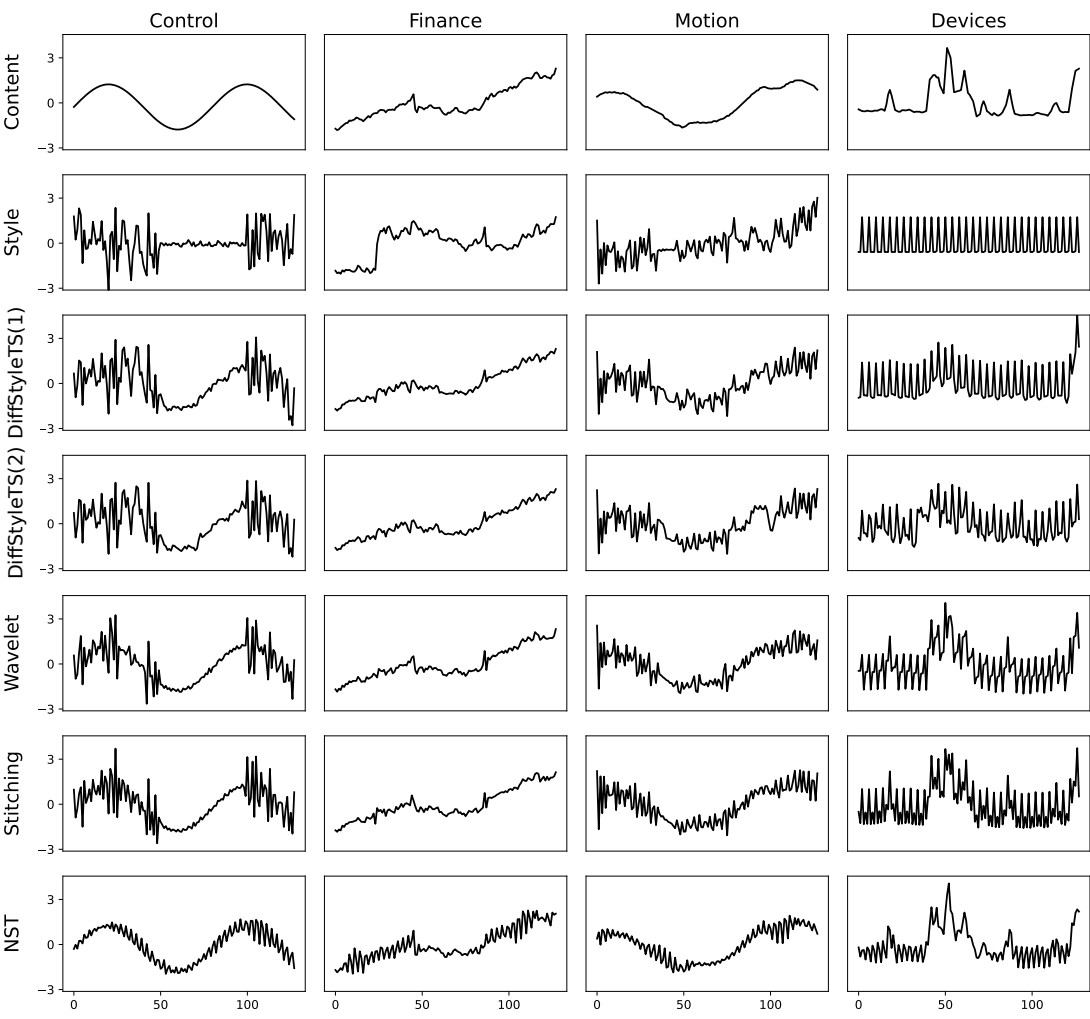

Figure 4: Qualitative comparison of style transfer methods across four content–style pairs (Control, Finance, Motion, Devices). Rows show the reference content, style, and outputs from DiffStyleTS (ours), wavelet, stitching, and neural optimization baselines. DiffStyleTS better integrates visible style dynamics into the generated series while preserving the underlying content trajectory; baseline methods tend to overfit to content or introduce artifacts.

while balancing content, style, and plausibility. Appendix E.2 shows that replacing the specialized encoders degrades style integration and realism, and Appendix E.3 shows how content and style guidance control the CP–SI trade-off during inference.

## 5.3 Qualitative Evaluation

We also evaluate DiffStyleTS qualitatively on unseen styles drawn from HIVE-COTE benchmark-family datasets (Middlehurst et al., 2021) excluded from training. Figure 4 compares DiffStyleTS with the baselines across four content–style pairs. Wavelets and Stitching mostly superimpose superficial fluctuations from the style series onto the content series, so their outputs remain content-dominant. For example, the abrupt jumps in the Control style series around timesteps 25–40 are missed by all baselines but reproduced by DiffStyleTS; in the Devices example, oscillatory spikes cause Wavelets and Stitching to introduce compensatory artifacts.

Table 2: Anomaly detection on NoBOOM with and without style-transfer augmentation. Scores are normalized ALARM scores from the revised pipeline with optimized window sizes. Higher is better.

| Training dataset | EIF | NeuTraLAD | LSTM-AE | FedFormer | GDN |
|---|---|---|---|---|---|
| OME | 26 | 50 | 33 | 47 | 20 |
| OME-ST | **28** | **52** | **60** | **57** | **37** |

The NST baseline preserves content but repeatedly inserts similar motifs, indicating limited sensitivity to style-specific local dynamics.

DiffStyleTS reproduces salient style dynamics while maintaining the content trajectory. The two DiffStyleTS samples also show that the same content–style pair can yield multiple plausible outputs, which is important when augmentation should expose downstream models to a range of realistic local variations. Appendix E.1 further shows how the temperature parameter $\lambda$ controls this diversity.

### 5.4 Simulation-to-Experiment Augmentation for Anomaly Detection in Chemical Data

We next test whether DiffStyleTS can improve low-data anomaly detection in chemical processes, where missed faults can cause off-spec products, equipment damage, downtime, or hazardous operating conditions. Detectors need normal-operation data that reflect real deployment variability, but only limited real data may be available. Simulators provide physically plausible process trajectories, but often miss the measurement noise, local variability, and operating dynamics of real experiments. TSST addresses this gap by transferring experimental style onto simulated trajectories while preserving process evolution. We evaluate this setting on NoBOOM, a real chemical-process anomaly-detection benchmark with phase-wise anomaly labels and a plant-operation evaluation protocol (Wagner et al., 2025). We focus on the OME process, which has only five normal training experiments, and report normalized ALARM scores in $[0, 100]$, where higher scores reward event detection, earlier detection within anomaly phases, and fewer false alarms (Wagner et al., 2026).

We construct *OME-ST*, an augmented OME training set built from style-transferred simulations. Smooth OME process-simulation trajectories provide content, while normal real OME training experiments provide style. DiffStyleTS converts these simulations into realistic training sequences with measurement noise, local variability, and operating dynamics. We append these generated sequences only to the training split; validation and test data remain real. OME and OME-ST use the same preprocessing, model classes, validation split, window-size search space, and hyperparameter-selection protocol; only the training data differ. Appendix I provides details on OME-ST construction, preprocessing, model configurations, and the evaluation protocol.

Table 2 compares detectors trained on OME with detectors trained on OME-ST. We evaluate EIF (Hariri et al., 2021), NeuTraLAD (Qiu et al., 2021), LSTM-AE (Malhotra et al., 2016), FedFormer (Zhou et al., 2022), and GDN (Deng & Hooi, 2021). OME-ST improves all reported baselines, with the largest gains for the deep models LSTM-AE, FedFormer, and GDN. This shows that style-transferred simulations can provide useful training data when real normal-operation data are scarce.

## 6 Conclusion

Generalizing time-series models to new operating conditions requires realistic variation beyond the observed data. DiffStyleTS addresses this need through time series style transfer: a single diffusion model, trained without paired examples, can preserve the low-frequency trajectory of one series while transferring high-frequency local dynamics from another. Across multi-domain qualitative and quantitative benchmarks, DiffStyleTS demonstrates effective style transfer. In the simulation-to-experiment study, DiffStyleTS improves anomaly detection in a low-data chemical-process setting by augmenting training data with more realistic simulations. Together, these results show that decomposing diffusion guidance into separate content and style controls enables flexible generation from unpaired time-series data and provides a foundation for extending guided generation to finer temporal structure, additional control axes, and multimodal settings.

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

## A    Model Architecture

This appendix specifies the implementation of the DiffStyleTS architecture used for all results. The model is implemented in a single PyTorch module, `DiffusionTransformer`.

**I/O conventions.**    All series are univariate tensors of shape $(B, 1, L)$. Training uses fixed windows of $L = 128$. When $L$ is not divisible by the patch size $p$, we right-pad to a multiple of $p$ and crop the output back to length $L$.

### Content encoder: low-pass, alignment-preserving

The content encoder $\mathcal{E}_\phi$ removes high-frequency detail while preserving temporal alignment:

- **Downsample:** 1D convolution with stride ds $= 8$, kernel $k = 5$, GELU.
- **Low-res trunk:** blocks $= 3$ conv blocks, each `Conv1d(H,H,k=5)` $\rightarrow$ GELU $\rightarrow$ `Conv1d(H,H,k=5)` $\rightarrow$ GELU, with $H = 128$ channels.
- **Project & upsample:** $1 \times 1$ `Conv1d` to a single channel, then linear interpolation back to the padded input length, followed by cropping to $L$.

The output $x^c \in \mathbb{R}^{B \times 1 \times L}$ is a smoothed, aligned representation.

### Style encoder: local, linear-phase, zero-DC

The style encoder $\mathcal{E}_\psi$ captures local high-frequency structure using small, linear-phase filters that cannot reintroduce offsets or phase shifts. Each `SymDCFreeConv1d` layer enforces (i) zero DC by subtracting the kernel mean and (ii) symmetry via averaging with the time-reversed kernel; reflection padding avoids edge artifacts. We use:

$$[\texttt{SymDCFreeConv1d(1,16,k=3)} \rightarrow \text{GELU}] \times 2 \rightarrow \texttt{1×1 Conv1d(16,1)}.$$

The output $x^s \in \mathbb{R}^{B \times 1 \times L}$ emphasizes high-frequency local dynamics.

### Patchification and unpatching

We use non-overlapping patches of length $p = 8$.

- **PatchEmbed:** linear projection from each flattened patch to $d$-dim tokens ($d = 256$), producing three token streams: noisy input $E_t$, content $E^c$, style $E^s$.
- **Unpatch:** linear projection back to patch space, reshape to $(B, 1, L)$, crop to original length.

### Time conditioning

The diffusion step $t$ is encoded with a $d$-dimensional sinusoidal embedding followed by a two-layer MLP with SiLU. The resulting vector is added to every token in $E_t$.

**Denoiser: stacked self/cross-attention with ALiBi**

The backbone comprises num_layers $= 4$ identical *DenoisingBlocks*. Each block applies:

1. self-attention over $E_t$ with ALiBi (additive linear positional biases);

2. cross-attention to content tokens $E^c$ (with ALiBi cross-bias);

3. cross-attention to style tokens $E^s$ (with ALiBi cross-bias);

4. position-wise MLP.

All sublayers use pre-LayerNorm and residual connections. We use num_heads $= 4$ and hidden size $d = 256$.

**Classifier-free guidance via input dropout**

During training, we randomly drop the *raw* conditioning streams: with probabilities $p_c = 0.10$ (content) and $p_s = 0.15$ (style), the respective input is replaced by zeros. The corresponding encoded features are also zeroed to prevent leakage. At inference, we obtain unconditional, content-only, and style-only predictions in a single forward pass and combine them with guidance scales $s_c$ and $s_s$ (Algorithm 2).

**Default configuration.** Unless stated otherwise: $d = 256$, $p = 8$, num_layers $= 4$, num_heads $= 4$, content encoder ($\text{ds} = 8, H = 128, \text{blocks} = 3, k = 5$), style encoder ($\text{hidden} = 16, \text{depth} = 2, k = 3$). With patch size $p$, the number of tokens is $N = \lceil \frac{L}{p} \rceil$ after padding; ALiBi bias tensors are constructed from $N$ (or $N_c, N_s$) and broadcast across heads and batch.

# B  Training Procedure

**Datasets and windowing.** We train on univariate series from Monash TSF via `datasets` (the subset names are listed in `run.py`). Cached raw targets are used; multivariate targets are split into univariate series. Sequences shorter than 128 are discarded. A batch is formed by sampling a series and a random contiguous window of length $L = 128$, and z-normalizing the window: $x' = (x - \mu)/(\sigma + 10^{-8})$. Missing values are replaced with zeros prior to normalization using `np.nan_to_num`.

**Self-supervised conditioning.** Each iteration uses the same normalized window as source for noisy input, content, and style ($x_0 = x_c = x_s$), enabling self-supervised training without paired data.

**Forward diffusion.** We use $T = 500$ steps with linear schedule $\beta_t \in [10^{-4}, 0.02]$, $\bar{\alpha}_t = \prod_{s \leq t}(1 - \beta_s)$. For $t \sim \{1, \ldots, T\}$ and $\epsilon \sim \mathcal{N}(0, I)$,

$$x_t = \sqrt{\bar{\alpha}_t}\, x_0 + \sqrt{1 - \bar{\alpha}_t}\, \epsilon.$$

**Objective and optimization.** The model predicts $\hat{\epsilon} = \epsilon_\theta(x_t, t, x_c, x_s)$ and is trained with MSE: $\mathcal{L} = \|\epsilon - \hat{\epsilon}\|_2^2$. We use AdamW (lr $3 \cdot 10^{-4}$). Unless noted, batch size is 256 and training runs for 200,000 iterations. Classifier-free conditioning uses the input-drop probabilities above.

**Logging and checkpoints.** Every 10 steps we append the loss to a CSV log; every `log_interval` steps we save a checkpoint containing iteration number, model state, and optimizer state. Training resumes from the latest checkpoint if present.

**Inference**

We generate with DDPM sampling for $T$ steps. At each step we compute $\epsilon_u$ (uncond.), $\epsilon_c$ (content-only), and $\epsilon_s$ (style-only) in a single forward pass by stacking inputs, and combine them

$$\epsilon = \epsilon_u + s_c(\epsilon_c - \epsilon_u) + s_s(\epsilon_s - \epsilon_u),$$

then apply the standard DDPM update with posterior variance. A temperature parameter $\lambda$ scales the injected Gaussian noise. Content and style windows are normalized independently; the final output is denormalized with the content statistics.

## C  Hyperparameters

Table 3: Default hyperparameters for DiffStyleTS.

| Architecture | |
|---|---|
| Hidden size ($d$) | 256 |
| Attention heads | 4 |
| Denoising blocks | 4 |
| Patch size ($p$) | 8 |
| Time embedding | sinusoidal($d$) $\to$ MLP (SiLU) |
| Content encoder | stride-8 Conv1d ($k = 5$), 3 conv blocks, $1 \times 1$ Conv1d |
| Style encoder | $2 \times$ [SymDCFreeConv1d ($k = 3$), GELU] $\to 1 \times 1$ Conv1d |
| Positional bias | ALiBi (self and cross) |
| CFG drop probs | $p_c = 0.10$ (content), $p_s = 0.15$ (style) |
| **Diffusion** | |
| Steps $T$ | 500 |
| $\beta$ schedule | linear from $10^{-4}$ to $2 \cdot 10^{-2}$ |
| Objective | MSE on added noise |
| **Training** | |
| Iterations | 200,000 |
| Batch size | 256 |
| Optimizer | AdamW (lr $3 \cdot 10^{-4}$) |
| Window length $L$ | 128 |
| Normalization | per-window z-score; NaNs $\to$ 0 before norm |
| Datasets | Monash TSF subsets (listed in `run.py`) |
| Checkpointing | save every `log_interval` steps; CSV loss log |
| **Inference** | |
| Guidance scales | $s_c = 1.0$, $s_s = 1.0$ (unless noted) |
| Temperature | $\lambda = 1.0$ |
| Length constraint | $L$ multiple of patch size ($p = 8$) |

Table 3 summarizes the default DiffStyleTS hyperparameters. The architecture is deliberately lightweight: a hidden size of 256, four attention heads, and four denoising blocks balance expressiveness with computational cost. The encoders are specialized for the TSST decomposition. The content encoder uses strided convolutions to capture coarse temporal structure, while the style encoder uses symmetry-preserving convolutions to extract finer-scale variation. ALiBi positional biases couple these streams and support varying sequence lengths.

The diffusion process uses 500 steps with a linear $\beta$ schedule, giving a practical trade-off between training stability and sampling speed. Training uses AdamW and a batch size of 256 for 200,000 iterations. Each window is z-score normalized, and NaNs are replaced with zeros before normalization.

During inference, the default guidance scales for content and style are both one. The temperature parameter controls stochasticity in generation; the diversity ablation below shows how varying $\lambda$ changes sample dispersion. Together, these choices provide a reproducible default configuration that balances fidelity, diversity, and efficiency across datasets.

## D   Evaluation Metrics

We evaluate style transfer with *content preservation* (CP), *style integration* (SI), and *realism* (RM), matching the requirements in Section 3. All metrics are computed on z-normalized windows of equal length. The decomposition below is a fixed evaluation operator, independent of DiffStyleTS training, and is applied identically to all methods.

**Multi-scale content/style decomposition**

Given $x \in \mathbb{R}^L$, apply averaging filters with kernels $K = (3, 5, 15)$ to obtain a smoothed content signal $C(x)$ and residual style signal $S(x)$. This follows the standard trend–remainder view of time-series decomposition (Cleveland et al., 1990): the smoothed component captures the coarse trajectory, while the residual captures local deviations. Let $x_{\text{smooth}}^{(0)} := x$ and, for $i = 1, 2, 3$,

$$x_{\text{smooth}}^{(i)} = \text{AvgFilter}_{K_i}\big(x_{\text{smooth}}^{(i-1)}\big),$$
$$x_{\text{res}}^{(i)} = x_{\text{smooth}}^{(i-1)} - x_{\text{smooth}}^{(i)}.$$

Set $C(x) := x_{\text{smooth}}^{(3)}$ and $S(x) := \sum_{i=1}^{3} x_{\text{res}}^{(i)}$.

**Metrics**

Let $\hat{x}$ be the generated series for content $a$ and style $b$ (all of length $L$):

$$\text{CP}(\hat{x}, a) = \tfrac{1}{L} \sum_{t=1}^{L} \big(C(\hat{x})_t - C(a)_t\big)^2,$$
$$\text{SI}(\hat{x}, b) = \tfrac{1}{L} \sum_{t=1}^{L} \big(S(\hat{x})_t - S(b)_t\big)^2.$$

CP therefore measures low-frequency agreement with the content reference, while SI measures residual agreement with the style reference. For realism, we embed content and style components with a frozen time-series foundation model $f(\cdot)$ and compute

$$\text{RM}(\hat{x}, a, b) = \tfrac{1}{2}\Big[\text{MSE}\big(f(C(\hat{x})), f(C(a))\big) + \text{MSE}\big(f(S(\hat{x})), f(S(b))\big)\Big].$$

We report per-dataset means and standard errors across test pairs, and the overall mean of CP, SI, and RM per sample (lower is better for all).

## E   Ablations

The ablations examine three aspects of DiffStyleTS: the specialized architecture, the diversity of the generated output, and the effect of content and style guidance parameters.

### E.1   Diversity

To analyze the effect of sampling temperature, we generate 20 style-transferred series from the same content–style pair while sweeping the temperature from 0 to 1. Figure 5 projects the resulting embeddings onto the first two PCA components. At low temperatures, the samples cluster tightly, indicating conservative modifications that preserve the original dynamics. As the temperature increases, the points spread farther along the PCA axes, reflecting stronger perturbations and greater diversity. This trend shows that DiffStyleTS can balance realism and variability: higher temperatures introduce more stylistic variation, while the generated series remain coherent in the embedding space of real data.

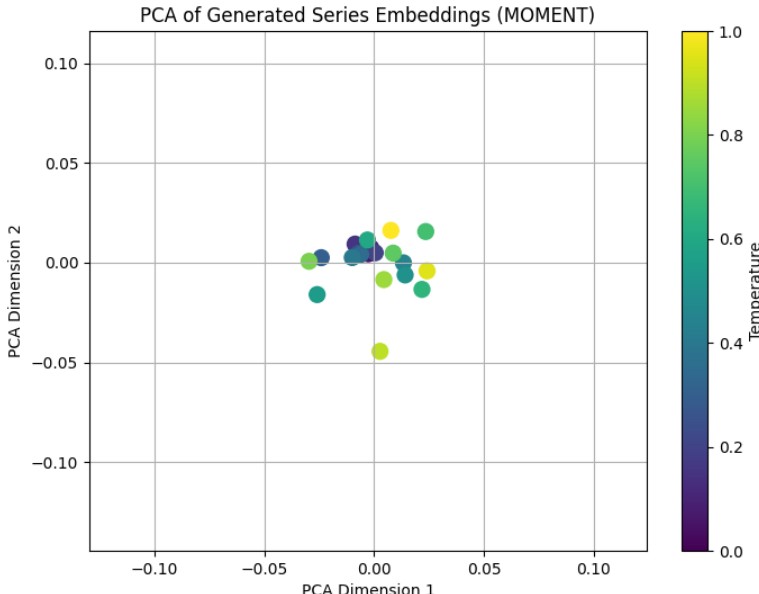

Figure 5: PCA of embeddings of style-transferred series at different temperatures. Higher temperatures lead to greater dispersion along the PCA axes, indicating increased diversity in the generated outputs while remaining within the manifold of realistic series.

Table 4: Architectural ablations of DiffStyleTS with convolutional replacements. To test the effect of specialized encoders, we replace them with plain Conv1d layers (kernel size 3), thereby removing the inductive biases for disentangling content and style while keeping model capacity comparable. Replacing the style encoder degrades style integration (SI), as the model defaults to extracting variability from the content stream, while content preservation (CP) remains strong. Replacing both encoders with Conv1d layers further harms all metrics, especially SI and realism (RM), confirming that inductive biases in the specialized encoders are essential for effective disentanglement and high-quality style transfer.

| Architecture Variant | CP $\downarrow$ | SI $\downarrow$ | RM $\downarrow$ | Avg $\downarrow$ |
|---|---|---|---|---|
| DiffStyleTS (full, specialized encoders) | 0.06 | 0.19 | 0.04 | 0.10 |
| Style Encoder $\to$ Conv1d($k$=3) | 0.02 | 0.36 | 0.06 | 0.15 |
| Content & Style Encoders $\to$ Conv1d($k$=3) | 0.26 | 0.65 | 0.12 | 0.34 |

### E.2 Architectural Ablations

To isolate the contribution of the specialized content and style encoders in DiffStyleTS, we replace them with generic convolutional layers of kernel size 3. This preserves model capacity but removes the carefully designed inductive biases: the content encoder's low-pass filtering for global structure, and the style encoder's zero-DC, symmetry-preserving convolutions for high-frequency variability. Table 4 reports the results.

**Full DiffStyleTS.** The complete model with both specialized encoders achieves the best overall balance across content preservation (CP), style integration (SI), and realism (RM). This confirms that explicitly disentangling content and style through tailored encoders provides the right inductive bias for the task.

Table 5: Ablation of classifier-free guidance scales. $s_c$ steers content adherence; $s_s$ steers style strength. Lower is better for all metrics; the best Avg occurs at $s_c = s_s = 1.0$.

| $s_c$ | $s_s$ | CP ↓ | SI ↓ | RM ↓ | Avg ↓ |
|---|---|---|---|---|---|
| 1.00 | 0.25 | 0.0653 | 1.0933 | 0.1133 | 0.4240 |
| 0.25 | 1.00 | 0.4906 | 0.0469 | 0.0339 | 0.1905 |
| 0.25 | 0.25 | 0.4299 | 0.9164 | 0.0994 | 0.4819 |
| 0.50 | 0.50 | 0.1065 | 0.6837 | 0.0878 | 0.2927 |
| 0.75 | 0.75 | 0.0643 | 0.4786 | 0.0587 | 0.2006 |
| 1.00 | 1.00 | 0.1283 | 0.0780 | 0.0191 | **0.0751** |
| 1.25 | 1.25 | 0.2207 | 0.0772 | 0.0184 | 0.1054 |
| 1.50 | 1.50 | 0.2465 | 0.1519 | 0.0281 | 0.1421 |
| 1.75 | 1.75 | 0.2862 | 0.2753 | 0.0313 | 0.1976 |
| 2.00 | 2.00 | 0.2596 | 0.6984 | 0.0392 | 0.3324 |

**Style encoder → Conv1d.** When the style encoder is replaced with a plain Conv1d layer, the model can no longer isolate high-frequency variability. As a result, SI worsens considerably, while CP appears to improve because the model primarily relies on the content stream. This shows that disentangling style information requires more than generic local filters.

**Content & style encoders → Conv1d.** When both encoders are replaced with Conv1d layers, the model loses nearly all disentanglement ability: CP, SI, and RM all deteriorate. Without inductive biases, information from content and style streams is blended during training, leading to leakage and poor controllability.

**Conclusion.** These ablations demonstrate that the inductive biases encoded in the specialized content and style encoders are crucial. They not only improve SI without sacrificing CP but also stabilize realism, leading to effective and controllable time-series style transfer.

### E.3 Content and Style Guidance

We study how classifier-free guidance scales affect the trade-off between content preservation (CP), style integration (SI), and realism (RM). Table 5 varies the content guidance $s_c$ and style guidance $s_s$.

**Observations.** (1) *Trade-off axis.* Skewed settings emphasize one objective at the expense of the other: high $s_c$, low $s_s$ (row 1.00/0.25) yields strong content adherence (low CP) but weak style integration (high SI), while low $s_c$, high $s_s$ (row 0.25/1.00) flips this behavior. (2) *Balanced sweet spot.* Equal, moderate guidance performs best overall: $s_c = s_s = 1.0$ minimizes the average score. Slightly stronger guidance (1.25/1.25) is competitive, but pushing both higher degrades CP and/or SI. (3) *Practical tuning.* For content-critical use, choose $s_c \gtrsim 1$, $s_s \lesssim 1$; for style-critical augmentation, invert the ratio. Use the sampling temperature $\lambda$ to modulate diversity *orthogonally* to guidance (Figure 5): increasing $\lambda$ spreads samples in PCA space without necessarily changing the CP/SI balance.

## F  Extended Quantitative Evaluation

**Reading the tables.** We report means ± standard errors across test pairs. Lower is better for all metrics. CP measures adherence to the content trajectory, SI measures transfer of local residual behavior, and RM checks representation-level plausibility. Deterministic baselines (Stitching, Wavelets) often *reuse* parts of the inputs, which favors content preservation but limits diversity; DiffStyleTS is generative and can trade content adherence for better style integration and realism.

**Content preservation.** As expected, Stitching, Wavelets, and NST achieve near-zero CP on most datasets because they either directly overlay residuals or optimize to the reference content. DiffStyleTS shows a small

but consistent CP penalty overall ($0.06 \pm 0.01$) because it generates a new sample with limited direct content copying. The largest CP gaps appear on datasets with strong piecewise-constant structure or long plateaus (*London Smart Meters* 0.19, *NN5 Daily* 0.18, *US Births* 0.26, *Vehicle Trips* 0.19, *Temperature (rain)* 0.17), where strict alignment to low-frequency level shifts is particularly hard without sacrificing style. This is a tunable trade-off: increasing the content guidance $s_c$, decreasing the style guidance $s_s$, or lowering the sampling temperature $\lambda$ (Algorithm 2) improves CP at the cost of style strength and diversity.

**Style integration.** DiffStyleTS is best (or tied) on the majority of datasets and by a clear margin overall ($0.19 \pm 0.04$ vs. $0.31/0.40/0.62$). Gains are pronounced where local volatility patterns matter: *NN5 Daily* (0.15 vs. 0.50/0.70/1.28), *US Births* (0.16 vs. 0.51/0.78/1.14), *Vehicle Trips* (0.13 vs. 0.48/0.70/1.07), *KDD Cup 2018* (0.20 vs. 0.30/0.36/0.55), and *Pedestrian Counts* (0.23 vs. 0.41/0.50/0.71). On very high-frequency signals (*Solar (4sec)*, *Wind (4sec)*) the gap narrows, and methods are effectively tied, indicating that simple residual swapping is sometimes adequate when style is dominated by white-like noise. Nevertheless, DiffStyleTS remains competitive even on challenging settings with mixed temporal scales (*Traffic (hourly)*: 0.51, the best among methods though absolute errors remain high for all). Because SI is computed from the fixed residual operator, these gains are independent of the MOMENT representation used for RM.

**Realism.** All methods achieve similar RM overall (Stitching 0.04, Wavelets 0.05, NST 0.05, DiffStyleTS 0.04). Despite not copying content, DiffStyleTS matches the best baselines on average and is often tied within rounding on individual datasets (e.g., *London Smart Meters*, *Tourism (monthly)*, *Weather*). Stitching sometimes has a slight edge on domains where simple residual recombination stays close to the training manifold (*Rideshare*), but DiffStyleTS frequently attains the lowest RM as well (*Bitcoin*, *KDD Cup 2018*, *Solar (4sec)*, *Wind (4sec)*). The parity in RM, together with superior SI, indicates that the generated series remain statistically plausible while achieving stronger stylistic fidelity.

**Takeaways.** (1) DiffStyleTS delivers substantially better style integration without sacrificing realism. (2) A modest CP gap reflects the difference between generating a new sample and copying content segments, and can be reduced by adjusting guidance $(s_c, s_s)$ and temperature $\lambda$. (3) On purely high-frequency styles, simple baselines can approach DiffStyleTS, but on mixed-scale and structured styles, diffusion with content/style conditioning provides clear advantages.

## G Zero-Shot Length Extrapolation with ALiBi

**Setup.** We train the denoiser on fixed windows of length $L{=}128$ using Attention with Linear Biases (ALiBi), which adds a linear distance penalty to attention scores and removes the need for learned absolute position embeddings. To test length extrapolation, we keep model weights and all sampling hyperparameters fixed and run inference at $L \in \{128, 256, 512, 1024, 2048\}$. Following the standard TSST pipeline, we condition on a smooth low-frequency content sequence and a high-frequency style sequence; both are synthetic controls used to isolate length scaling. Sampling uses $s_c{=}s_s{=}1.0$ and temperature 0.0 for all $L$.

**Results.** Table 9 lists the quantitative values that appear in Figure 6. Across contexts up to $16\times$ longer than the training window, all metrics remain in a comparable range. CP improves slightly up to $L{=}1024$ and rebounds mildly at $L{=}2048$, while SI remains stable enough to indicate that high-frequency fluctuations are injected across the generated horizon. RM stays low across all $L$, indicating no obvious realism degradation due to extrapolation.

## H Code Availability

We include a code archive with the submission to support transparency and reuse of the proposed method. The archive contains the implementation of DiffStyleTS, including the model definition, training procedure, inference pipeline, and a pretrained checkpoint for running style transfer directly. The provided README describes the environment setup, expected input format, training command, inference command, and default hyperparameters.

Table 6: Evaluation (Content Preservation (CP)). Lower is better.

| Dataset | Stitching | Wavelets | NST | DiffStyleTS |
|---|---|---|---|---|
| Electricity Demand | 0.00 ± 0.00 | 0.00 ± 0.00 | **0.00 ± 0.00** | 0.00 ± 0.00 |
| Bitcoin | 0.00 ± 0.00 | **0.00 ± 0.00** | 0.00 ± 0.00 | 0.01 ± 0.00 |
| Covid Deaths | 0.00 ± 0.00 | 0.00 ± 0.00 | **0.00 ± 0.00** | 0.01 ± 0.00 |
| FRED-MD | 0.00 ± 0.00 | **0.00 ± 0.00** | 0.00 ± 0.00 | 0.00 ± 0.00 |
| Kaggle Web Traffic | 0.00 ± 0.00 | **0.00 ± 0.00** | 0.00 ± 0.00 | 0.01 ± 0.00 |
| KDD Cup 2018 | 0.00 ± 0.00 | **0.00 ± 0.00** | 0.00 ± 0.00 | 0.02 ± 0.00 |
| London Smart Meters | 0.00 ± 0.00 | 0.00 ± 0.00 | **0.00 ± 0.00** | 0.19 ± 0.00 |
| NN5 Daily | 0.00 ± 0.00 | **0.00 ± 0.00** | 0.00 ± 0.00 | 0.18 ± 0.00 |
| Oikolab Weather | 0.01 ± 0.00 | 0.00 ± 0.00 | **0.00 ± 0.00** | 0.06 ± 0.00 |
| Pedestrian Counts | 0.00 ± 0.00 | **0.00 ± 0.00** | 0.00 ± 0.00 | 0.03 ± 0.00 |
| Rideshare | 0.00 ± 0.00 | **0.00 ± 0.00** | 0.00 ± 0.00 | 0.03 ± 0.00 |
| Saugeenday | 0.00 ± 0.00 | **0.00 ± 0.00** | 0.00 ± 0.00 | 0.01 ± 0.00 |
| Solar (10min) | 0.00 ± 0.00 | 0.00 ± 0.00 | **0.00 ± 0.00** | 0.00 ± 0.00 |
| Solar (4sec) | 0.00 ± 0.00 | **0.00 ± 0.00** | 0.00 ± 0.00 | 0.00 ± 0.00 |
| Sunspot | 0.00 ± 0.00 | **0.00 ± 0.00** | 0.00 ± 0.00 | 0.05 ± 0.00 |
| Temperature (rain) | 0.00 ± 0.00 | **0.00 ± 0.00** | 0.00 ± 0.00 | 0.17 ± 0.00 |
| Tourism (monthly) | 0.00 ± 0.00 | 0.00 ± 0.00 | **0.00 ± 0.00** | 0.09 ± 0.00 |
| Traffic (hourly) | 0.00 ± 0.00 | 0.00 ± 0.00 | **0.00 ± 0.00** | 0.04 ± 0.00 |
| US Births | 0.00 ± 0.00 | 0.00 ± 0.00 | **0.00 ± 0.00** | 0.26 ± 0.00 |
| Vehicle Trips | 0.00 ± 0.00 | 0.00 ± 0.00 | **0.00 ± 0.00** | 0.19 ± 0.00 |
| Weather | 0.00 ± 0.00 | 0.00 ± 0.00 | **0.00 ± 0.00** | 0.08 ± 0.00 |
| Wind (4sec) | 0.00 ± 0.00 | 0.00 ± 0.00 | **0.00 ± 0.00** | 0.00 ± 0.00 |
| Wind Farms (min) | 0.00 ± 0.00 | **0.00 ± 0.00** | 0.00 ± 0.00 | 0.05 ± 0.01 |
| Overall | 0.00 ± 0.00 | 0.00 ± 0.00 | 0.00 ± 0.00 | 0.06 ± 0.01 |

The implementation follows the configuration reported in Table 3, including the specialized content and style encoders, ALiBi-based denoising backbone, classifier-free guidance, DDPM sampling procedure, and per-window normalization used throughout the experiments. The included checkpoint can be used to generate style-transferred time series without retraining, while the training code allows the model to be trained from scratch on the public time-series datasets described in Appendix B.

# I   Application: Anomaly Detection in Chemical Data

We evaluate the downstream utility of DiffStyleTS on the OME subset of NoBOOM (Wagner et al., 2025), a real-world benchmark for time-series anomaly detection in chemical processes. NoBOOM emphasizes early and reliable detection in operating plants: its ALARM score rewards detecting anomaly events, favors earlier detection within labeled anomaly phases, and penalizes excessive false alarms (Wagner et al., 2026).

OME is the continuous reactive distillation configuration. It contains five normal training sequences and three test sequences with four labeled anomaly events, corresponding to 986 test time steps and 36[ M101, M102, M103, PIC101, R, T101, T102, T104, T106, T108, T110, T112, T114. ] All anomaly detectors therefore receive 13-channel inputs.

**Style-transferred OME data.**   OME-ST denotes the augmented OME training set used in the downstream experiment. It consists of the original real OME training sequences together with style-transferred simulation sequences generated by DiffStyleTS. Process simulation data provides content, and normal experimental OME training data provides style. Since OME is multivariate while DiffStyleTS is univariate in this study, style transfer is applied separately to each retained process variable. For each channel, the simulated trajectory provides content and the corresponding real normal channel provides style; the generated channels are then recombined on the shared time grid to form multivariate training sequences for the anomaly detectors.

Table 7: Evaluation (Style Integration (SI)). Lower is better.

| Dataset | Stitching | Wavelets | NST | DiffStyleTS |
|---|---|---|---|---|
| Electricity Demand | 0.18 ± 0.06 | 0.20 ± 0.07 | 0.32 ± 0.08 | **0.18 ± 0.06** |
| Bitcoin | 0.14 ± 0.01 | 0.16 ± 0.02 | 0.27 ± 0.02 | **0.11 ± 0.02** |
| Covid Deaths | 0.19 ± 0.04 | 0.23 ± 0.06 | 0.34 ± 0.06 | **0.16 ± 0.03** |
| FRED-MD | 0.08 ± 0.01 | 0.09 ± 0.01 | 0.20 ± 0.01 | **0.08 ± 0.01** |
| Kaggle Web Traffic | 0.19 ± 0.01 | 0.22 ± 0.02 | 0.34 ± 0.02 | **0.12 ± 0.01** |
| KDD Cup 2018 | 0.30 ± 0.03 | 0.36 ± 0.03 | 0.55 ± 0.05 | **0.20 ± 0.01** |
| London Smart Meters | 0.42 ± 0.01 | 0.48 ± 0.01 | 0.83 ± 0.01 | **0.33 ± 0.03** |
| NN5 Daily | 0.50 ± 0.06 | 0.70 ± 0.09 | 1.28 ± 0.09 | **0.15 ± 0.02** |
| Oikolab Weather | 0.39 ± 0.05 | 0.52 ± 0.07 | 0.67 ± 0.13 | **0.26 ± 0.04** |
| Pedestrian Counts | 0.41 ± 0.01 | 0.50 ± 0.01 | 0.71 ± 0.01 | **0.23 ± 0.01** |
| Rideshare | 0.42 ± 0.10 | 0.51 ± 0.12 | 0.71 ± 0.15 | **0.34 ± 0.05** |
| Saugeenday | 0.23 ± 0.02 | 0.26 ± 0.02 | 0.39 ± 0.02 | **0.14 ± 0.01** |
| Solar (10min) | 0.17 ± 0.04 | 0.20 ± 0.04 | 0.30 ± 0.05 | **0.16 ± 0.05** |
| Solar (4sec) | **0.10 ± 0.02** | 0.11 ± 0.03 | 0.21 ± 0.03 | 0.10 ± 0.03 |
| Sunspot | 0.52 ± 0.08 | 0.62 ± 0.12 | 0.87 ± 0.18 | **0.29 ± 0.05** |
| Temperature (rain) | 0.47 ± 0.02 | 0.74 ± 0.02 | 1.18 ± 0.02 | **0.30 ± 0.04** |
| Tourism (monthly) | 0.30 ± 0.01 | 0.38 ± 0.01 | 0.83 ± 0.01 | **0.14 ± 0.04** |
| Traffic (hourly) | 0.69 ± 0.09 | 0.80 ± 0.10 | 0.93 ± 0.11 | **0.51 ± 0.07** |
| US Births | 0.51 ± 0.03 | 0.78 ± 0.03 | 1.14 ± 0.03 | **0.16 ± 0.05** |
| Vehicle Trips | 0.48 ± 0.03 | 0.70 ± 0.04 | 1.07 ± 0.04 | **0.13 ± 0.03** |
| Weather | 0.31 ± 0.01 | 0.43 ± 0.01 | 0.66 ± 0.01 | **0.18 ± 0.03** |
| Wind (4sec) | 0.12 ± 0.03 | 0.13 ± 0.04 | 0.23 ± 0.04 | **0.12 ± 0.03** |
| Wind Farms (min) | 0.08 ± 0.01 | 0.09 ± 0.01 | 0.16 ± 0.01 | **0.08 ± 0.01** |
| Overall | 0.31 ± 0.06 | 0.40 ± 0.10 | 0.62 ± 0.17 | 0.19 ± 0.04 |

Simulation trajectories provide smooth, physically plausible process evolution, while the experimental style references contribute measurement noise, local variability, and operating dynamics. The generated sequences are added only to the training split. Validation and test data remain real, so the reported gains measure whether OME-ST helps detectors generalize to real plant measurements.

**Simulation source.** The simulated content trajectories are based on the first-principles mini-plant modeling approach described by Ferre et al. (2024), which represents the main process units separately rather than as a single black-box model. The reaction system accounts for reversible formation of formaldehyde oligomers with water and methanol, while OME formation under acidic conditions is modeled kinetically. Vapor–liquid equilibrium is computed with extended Raoult's law and a non-ideal liquid-phase activity model, and possible solid formation is checked through a separate solid–liquid equilibrium model. The reactor is represented as an isothermal plug-flow reactor, whereas the distillation column is simulated with an Aspen Plus equilibrium-stage model augmented by custom routines for reaction and phase-equilibrium calculations. Ferre et al. (2024) compare the resulting simulations with mini-plant composition and temperature profiles to assess reactor behavior, separation performance, product purity, and precipitation risk; in OME-ST, these smooth simulation trajectories provide the process-consistent content that is stylized with normal experimental measurements.

**Data splits and preprocessing.** The OME-ST construction reuses the base `cont_reactive_ome` data, intersects the real and style-transferred feature sets, and appends seven style-transferred training sequences to the five real training sequences. Style-transfer content trajectories and real style references are selected only from training data; real validation and test sequences are excluded from OME-ST construction. The final split sizes are: OME training, 5 sequences / 2387 steps; style-transferred training addition, 7 sequences / 3752 steps; real validation, 5 sequences / 1020 steps; and real test, 3 sequences / 986 steps. We use a

Table 8: Evaluation (Realism (RM)). Lower is better.

| Dataset | Stitching | Wavelets | NST | DiffStyleTS |
|---|---|---|---|---|
| Electricity Demand | **0.03 ± 0.00** | 0.04 ± 0.00 | 0.05 ± 0.00 | 0.03 ± 0.00 |
| Bitcoin | 0.04 ± 0.00 | 0.04 ± 0.00 | 0.06 ± 0.00 | **0.03 ± 0.00** |
| Covid Deaths | 0.03 ± 0.00 | 0.05 ± 0.00 | 0.06 ± 0.00 | **0.03 ± 0.00** |
| FRED-MD | 0.04 ± 0.00 | 0.05 ± 0.00 | 0.06 ± 0.00 | **0.03 ± 0.00** |
| Kaggle Web Traffic | 0.04 ± 0.00 | 0.04 ± 0.00 | 0.05 ± 0.00 | **0.03 ± 0.00** |
| KDD Cup 2018 | 0.03 ± 0.00 | 0.03 ± 0.00 | 0.04 ± 0.00 | **0.03 ± 0.00** |
| London Smart Meters | 0.05 ± 0.00 | 0.04 ± 0.00 | **0.04 ± 0.00** | 0.04 ± 0.00 |
| NN5 Daily | 0.06 ± 0.00 | 0.07 ± 0.00 | 0.06 ± 0.00 | **0.04 ± 0.00** |
| Oikolab Weather | 0.03 ± 0.00 | 0.04 ± 0.00 | 0.04 ± 0.00 | **0.03 ± 0.00** |
| Pedestrian Counts | 0.06 ± 0.00 | 0.05 ± 0.00 | 0.05 ± 0.00 | **0.05 ± 0.00** |
| Rideshare | **0.04 ± 0.00** | 0.05 ± 0.00 | 0.05 ± 0.00 | 0.05 ± 0.00 |
| Saugeenday | **0.03 ± 0.00** | 0.03 ± 0.00 | 0.05 ± 0.00 | 0.04 ± 0.00 |
| Solar (10min) | 0.04 ± 0.00 | 0.05 ± 0.00 | 0.06 ± 0.00 | **0.03 ± 0.00** |
| Solar (4sec) | **0.03 ± 0.00** | 0.04 ± 0.00 | 0.07 ± 0.00 | 0.03 ± 0.00 |
| Sunspot | **0.03 ± 0.00** | 0.03 ± 0.00 | 0.03 ± 0.00 | 0.04 ± 0.00 |
| Temperature (rain) | 0.05 ± 0.00 | **0.05 ± 0.00** | 0.07 ± 0.00 | 0.05 ± 0.00 |
| Tourism (monthly) | 0.04 ± 0.00 | **0.04 ± 0.00** | 0.05 ± 0.00 | 0.05 ± 0.00 |
| Traffic (hourly) | **0.04 ± 0.00** | 0.04 ± 0.00 | 0.04 ± 0.00 | 0.05 ± 0.00 |
| US Births | 0.08 ± 0.00 | 0.07 ± 0.00 | 0.07 ± 0.00 | **0.05 ± 0.00** |
| Vehicle Trips | 0.08 ± 0.00 | 0.07 ± 0.00 | 0.07 ± 0.00 | **0.04 ± 0.00** |
| Weather | 0.04 ± 0.00 | **0.04 ± 0.00** | 0.04 ± 0.00 | 0.04 ± 0.00 |
| Wind (4sec) | **0.03 ± 0.00** | 0.04 ± 0.00 | 0.06 ± 0.00 | 0.03 ± 0.00 |
| Wind Farms (min) | 0.07 ± 0.00 | 0.17 ± 0.00 | 0.09 ± 0.00 | **0.07 ± 0.00** |
| Overall | 0.04 ± 0.00 | 0.05 ± 0.00 | 0.05 ± 0.00 | 0.04 ± 0.00 |

Table 9: Successful length extrapolation with ALiBi. Style transfer quality remains consistent across longer horizons. Metrics correspond to Figure 6.

| $L$ | CP ↓ | SI ↓ | RM ↓ |
|---|---|---|---|
| 128 | 0.134 | 0.138 | 0.023 |
| 256 | 0.128 | 0.143 | 0.009 |
| 512 | 0.115 | 0.176 | 0.022 |
| 1024 | 0.087 | 0.194 | 0.016 |
| 2048 | 0.119 | 0.195 | 0.027 |

mixed robust preprocessing pipeline with zero-inflation handling and asinh transformation enabled. Robust scaling is disabled. The preprocessing state is fit on the training export: real OME training data for the OME condition, and real OME training data plus the style-transferred addition for the OME-ST condition. During training export, extreme values are handled with forward fill; at test export, the fitted preprocessing state is reused and extreme handling is disabled.

**Evaluation protocol.** All models use sliding windows with stride 1. Only complete windows are emitted: no padding is used and incomplete trailing windows are dropped. The OME and OME-ST conditions use the same validation split, preprocessing choices, model families, and candidate window-size range; only the training data differ. Neural models are trained with AdamW, learning rate $10^{-4}$, maximum 100 epochs, validation every 3 epochs, early stopping patience 5, and minimum validation-loss improvement $10^{-4}$. Batch size is 128 except for NeuTraLAD, which uses batch size 64.

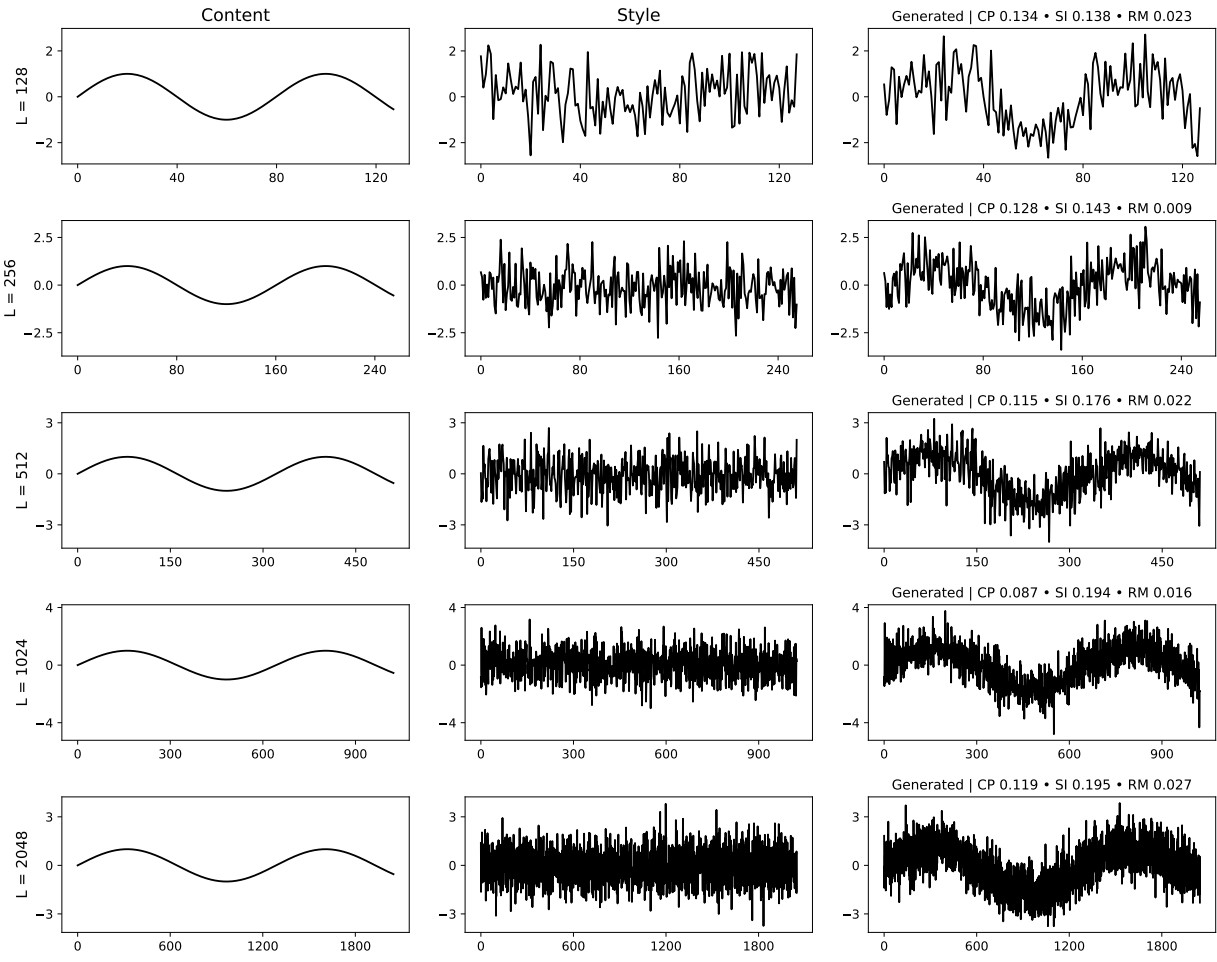

Figure 6: DiffStyleTS enables zero-shot length extrapolation with stable metrics. Rows show increasing sequence lengths $L \in \{128, 256, 512, 1024, 2048\}$; columns show the *content* input, the *style* input, and the *generated* output. The generated column annotates CP, SI, and RM for each $L$. The same model, trained only at $L$=128, preserves global content shape and consistently integrates high-frequency style across much longer horizons.

Table 10: Windowing configuration for the OME-ST experiment.

| Model | Window | Horizon | Treatment |
|---|---|---|---|
| EIF | 128 | – | window isolation score |
| GDN | 98 | 1 | one-step prediction |
| LSTM-AE | 88 | – | reconstruction with featurewise thresholding |
| FedFormer | 88 | – | reconstruction with featurewise thresholding |
| NeuTraLAD | 118 | – | reconstruction; test-on-original enabled |

**Results and interpretation.** Table 2 reports the optimized-window comparison from the main text. OME-ST improves EIF, NeuTraLAD, LSTM-AE, FedFormer, and GDN. This pattern is consistent with the intended role of DiffStyleTS: converting simulations into additional realistic normal-operation training data.

Table 11: Model settings for the OME-ST experiment.

| Model | Selected architecture / hyperparameters |
|---|---|
| EIF | 100 trees, sample size 128, all feature combinations, 4 CPU jobs |
| GDN | latent dimension 64, top-$k = 15$, output hidden layers $[64, 64]$, dropout 0.1 |
| LSTM-AE | hidden layers $[50, 50]$, dropout 0.5 |
| FedFormer | model dimension 480, 8 heads, 3 encoder layers, 48 modes, dropout 0.4, top-$k = 24$, shared attention and scattered frequencies enabled |
| NeuTraLAD | 4 transformations, forward type, encoder hidden 128, 1 encoder layer, 1 transform layer, latent 64, batch normalization, PatchTST encoder, MLP head |

