# OpenReview forum: "DiffStyleTS: Diffusion Model for Style Transfer in Time Series"
_TMLR — Under review for TMLR_

### Review · Reviewer_reep · 2026-06-22

**Summary Of Contributions:**

The paper brings together some of the well known techniques and applies them to the domain of Time Series. The authors use the ideas behind diffusion models and style transfer and present an idea of being able to do Time Series Style Transfer(TSST). In this regard the idea feels novel as well as interesting.

There is additional contribution in terms of how they break style transfer in terms of content and style and what it means from a time series perspective, how content and style maps to time series. Then they go on to present how they use convolution based architectures to separate out the content and the style.

All these are interesting ideas and novel contributions that build on top of existing work and definitely provide an interesting approach to the problem.

**Audience:**

Yes

**Audience Explanation:**

Yes, as mentioned above the authors build on top of some fundamental ideas like diffusion and style transfer and bring them over to time series, as well as instroduce a notion of content and style for time series. For folks in time series domain this will definitely be of interest.

**Claims And Evidence:**

Yes

**Claims Explanation:**

Yes, both the Time Series Style transfer approach, alongside with clear architecture on how to implement the approach and the fact that approach works as shown with benchmarks across a series of datasets all point in the direction that the approach works.

Authors never make the claim that this is state of the art approach, they have shown an interesting direction of work and have carried out experiments along that direction and presented the results. More needs to be done both in terms of throughness of the experiments and presentation of results to show the approach is of significant value, but as a novelty and an interesting approach the paper holds merit.

**Requested Changes:**

None, I am not an expert in this area to recommend any major changes.

---

### Review · Reviewer_dZ37 · 2026-07-09

**Summary Of Contributions:**

The submission concerns the application of diffusion-based style transfer to time series data, commonly employed as part of a data augmentation strategy. For video and text, the goal of style transfer is to re-represent the content from one signal in the style of another. In the context of time-series style transfer, the "content" is the low-frequency global trajectory and the "style" are the high frequency local dynamics. The architecture for the proposed approach uses separate encoders for style and transfer and stacked diffusion transformer blocks, adapted for time-series. Quantitative and qualitative evaluation across several timeseries domains using styles not seen during training support the superiority of the proposed method compared to three baseline methods. Further evaluation uses the proposed method for data augmentation in an anomaly detection task and finds that the style-transfer-augmented dataset leads to significant improvement in anomaly detection across five different detection methods.

**Strengths and weaknesses:**
The submission is well written and tackles a well-defined problem. Multiple evaluations are presented on diverse datasets. Figures and tables are generally clear but I have requested minor clarifications. The paper would benefit from an expanded discussion of limitations. No statistical tests are reported and there is limited discussion about whether the improvement reported here is meaningful. It would be more convincing if the anomaly detection evaluation compared competing style transfer methods for data augmentation.

**Audience:**

Yes

**Audience Explanation:**

This paper will be relevant to the timeseries style transfer community or anyone needing data augmentation for time series data.

**Broader Impact Concerns:**

Broader Impact Statement is not needed.

**Claims And Evidence:**

Yes

**Claims Explanation:**

Claims of the superiority of their method are supported by both quantitative and qualitative results. Claims that the approach can be useful for data augementation are supported by the anomaly detection results. The submission stops short of making the claim (although perhaps it is somewhat implied) that their method should be preferred for time-series data augmentation, which is not fully supported.

**Requested Changes:**

- The Conclusion is very short. There is no discussion of limitations. What hasn't been tested? Where does this method fail and why? Are there other pros and cons of the evaluated methods, beyond the scores presented here, e.g., model complexity or data hungriness? Should your method be preferred in all settings? Please discuss.
- The anomaly detection evaluation only compares the OME dataset with and without style-transfer augmentation. It would be stronger if you included other data augmentation approaches as well. Is it the case that any data augmentation would produce the benefits shown here? Or it is specific to your style transfer approach?
- Table 1 contains several ties. I assume that you have bolded the numerically smallest method, even when the differences are small enough to be lost in rounding. Please clarify in the caption for Table 1 your policy for bolding when there are ties. Consider bolding all methods that are statistically indistinguishable from the best.
- Table 1: Please be explicit in the figure caption about what the uncertainty +/- bounds refer to. I assume standard deviation but best to be explicit.
- Figure 4 has two rows for DiffStyleTS(1) and DiffStyleTS(2) but these are not explained in the caption. What are these two versions of your method? Please explain in the caption.
- Although it is commonplace to report a table of scores + standard deviation like in Table 1, it would be better to also include a statistical quantification of the effect or to otherwise provide context to interpret whether the differences reported are of consequence.

---

### Review · Reviewer_qhHA · 2026-07-18

**Summary Of Contributions:**

The paper proposes DiffStyleTS, a diffusion-based framework for content-style transfer. The framework has three components, adapted for time series data: content encoder, style encoder, and a diffusion transformer as the denoising network conditioned on both inputs. The goal of the framework is to generate a new time series datapointfrom two existing datapoints served as content and style.

**Audience:**

Yes

**Audience Explanation:**

The idea of style-content transfer for time-series data is interesting and novel.

However, I feel the problem formulation requires stronger support. In particular, the notions of content and style adopted in this work differ from those in prior work, yet the paper provides limited justification for why this formulation is appropriate. It almost feels that identifying an appropriate definition of content versus style is itself one of the central scientific challenges in tsst.

**Claims And Evidence:**

No

**Claims Explanation:**

There are a few remaining questions to be answered before I can provide a definitive answer.

- Could the authors please provide some justification for their definition of content and style? It was defined in the paper that content is low-frequency data while the style component is high frequency local dynamics. Should we also consider alternative definitions (such as distributional characteristics defined in StyleTime)?

- How can we be confident that the proposed evaluation metrics are sufficiently complete? The three proposed metrics (CP, SI, and RM) appear to be closely coupled with the paper’s definition of content and style. Consequently, the authors may want the show the results are not specific to the chosen formulation and evaluation protocol. Additional justification for the metrics, or complementary evaluations under alternative notions of content and style, would strengthen the paper.

**Requested Changes:**

See above. I think the topic is interesting and I am happy to recommend if the questions are appropriately addressed.